# Molecular subtypes of Adenovirus-associated acute respiratory infection outbreak in children in Northern Vietnam and risk factors of more severe cases

**Dinh-Dung Nguyen**[1], **Lan Tuyet Phung**[2,3], **Huyen Thi Thanh Tran**[1], **Ha Thi Thanh Ly**[1], **Anh Hang Mai Vo**[1], **Nhung Phuong Dinh**[1], **Phuong Mai Doan**[4], **Anh Thi Nguyen**[4], **Luc Danh Dang**[4], **Thia Thi Doan**[4], **Khuong Thi Pham**[2], **Huong Lan Pham**[2], **Dai Xuan Hoang**[2], **Thao Ngoc Pham**[3], **Bao Thai Tran**[3], **Trang Thi Thuc Tran**[3], **Huong Thi Minh Le**[2], **An Nhat Pham**[2], **Antony Antoniou**[5], **Nhan Thi Ho**[6]*

1 Medical Genetics Department, Vinmec High Tech Center, Vinmec Healthcare System, Hanoi, Vietnam, 2 Pediatric Center, Vinmec Times City International General Hospital, Vinmec Healthcare System, Hanoi, Vietnam, 3 VinUniversity, Hanoi, Vietnam, 4 Microbiology Lab, Laboratory Department, Vinmec Times City International General Hospital, Vinmec Healthcare System, Hanoi, Vietnam, 5 Department of Applied Sciences, Faculty of Health and Life Sciences, Northumbria University, Newcastle upon Tyne, United Kingdom, 6 Research & Development Department, Vinmec High Tech Center, Vinmec Healthcare System, Hanoi, Vietnam

* v.nhanht6@vinmec.com

## Abstract

### Background

Under the pressure of Human Adenovirus (HAdV)–associated acute respiratory infection (ARI) outbreak in children in Northern Vietnam in the end of 2022, this study was initiated to identify the HAdV subtype(s) and examine the associated clinical features and risk factors of more severe cases.

### Methods

This study evaluated pediatric patients with ARI which had tested positive for HAdV between October and November 2022 using a multiplex real-time PCR panel. Nasopharyngeal aspirates or nasal swab samples were used for sequencing to identify HAdV subtypes. Clinical data were collected retrospectively.

### Results

Among 97 successfully sequenced samples, the predominant subtypes were HAdV-B3 (83%), HAdV-B7 (16%) and HAdV-C2 (1%). Lower respiratory manifestations were found in 25% of the patients of which 5% were diagnosed with severe pneumonia. There was no significant association between HAdV subtype and clinical features except higher white blood cell and neutrophil counts in those detected with HAdV-B3 (p<0.001). Co-detection of HAdV with ≥1 other respiratory viruses was found in 13/24(54%) of those with lower respiratory manifestations and 4/5(80%) of those with severe pneumonia (odds ratio (95% confidence

**Data Availability Statement:** The authors confirm that the data supporting the findings of this study

are available within the article and its supplementary materials.

**Funding:** This work was supported by the Research & Development fund of Vinmec High-tech Centre, Vinmec Healthcare System, Hanoi, Vietnam (to DDN, LTTH, and HTTT). The funders had no role in study design, data collection and analysis, decision to publish, or preparation of the manuscript.

**Competing interests:** The authors have declared that no competing interests exist.

interval) vs. those without = 10.74 (2.83, 48.17) and 19.44 (2.12, 492.73) respectively after adjusting for age, sex, birth delivery method, day of disease).

## Conclusion

HAdV-B3 and HAdV-B7 were predominant in the outbreak. Co-detection of HAdV together with other respiratory viruses was a strong risk factor for lower respiratory tract illnesses and severe pneumonia. The findings advocate the advantages of multi-factor microbial panels for the diagnosis and prognosis of ARI in children.

### Author summary

Human Adenoviruses (HAdV) is among the common causes of acute respiratory infection (ARI) in children. HAdV may cause severe or deadly diseases in both immunocompetent and immunocompromised individuals. There are many HAdV subtypes and different HAdV subtypes may be associated with different diseases or disease manifestations. In the second half of 2022, for the first time in Northern Vietnam, there was a HAdV-associated ARI outbreak in children with an alarming number of severe cases and deaths. During the outbreak, the authors performed Sanger sequencing on nasal swab samples of 97 patients who tested positive for HAdV by multiplex real-time PCR panel (the PCR panel was designed to test for HAdV and 6 other respiratory viruses). Three HAdV subtypes were identified: B3 (83%), B7 (16%) and C2 (1%). Patient clinical characteristics were similar between the detected HAdV subtypes. Co-detection of HAdV with other respiratory viruses was associated with greater disease severity compared to HAdV detection alone. This is probably the first study describing the clinical features and HAdV molecular subtypes associated with an ARI outbreak in children in Vietnam. The findings demonstrate the usefulness of multi-factor microbial panels for the diagnosis and prognosis of ARI in children.

## Introduction

Human Adenoviruses (HAdV) are classified into seven known HAdV species HAdV-A to G. The HAdV species may be classified further into different serotypes by traditional methods such as hemagglutination and serum neutralization reactions or more advanced methods e.g., genomic sequencing and bioinformatic analysis [1]. More than a hundred serotypes of HAdV have been documented thus far. HAdV serotypes are now vetted by the Human Adenovirus Working Group, a collaboration between adenoviral researchers and the National Center for Biotechnology Information (NIH)/GenBank [2]. HAdV may cause many common diseases such as upper respiratory infection, conjunctivitis, and gastrointestinal disorders. In addition, HAdV may cause severe or fatal diseases in both immunocompetent and immunocompromised individuals [3,4].

There had been sporadic HAdV outbreaks with respiratory illnesses [5–9], conjunctivitis [10,11], and gastrointestinal disorders [12,13] worldwide before the COVID-19 pandemic. Respiratory illnesses were reported in most of these HAdV infection studies, with severe respiratory illnesses in children having been reported in some studies [6,8,10–12,14–31]. Some populations exhibited similar prevalent respiratory HAdV species in children. For instance, the prevalent respiratory HAdV species detected in Taiwan and China were HAdV-B3, -C2, and

-B7 [5,8,10,18,20,25–27], whilst in Egypt and Argentina HAdV-B7 and -B3 were prevalent [16,29,32]. Other studies of HAdV species infecting children include those of Japan: HAdV-B3, -C2, and -C1 [17], Spain: HAdV-B3, -C6, and -C5 [19], and Palestine: HAdV-C1, -C2, -B3 and -C5 [22]. HAdV-B7, -B3, and -C2 were the most common types which were associated with severe respiratory illnesses in children, especially in those with underlying diseases or immunocompromised conditions [6,10,26,27,29–31]. During the early stages of the COVID-19 pandemic with social distancing until 2021, respiratory tract infections in children including HAdV infections exhibited a remarkable decrease in occurrence [9,28,33,34].

In Vietnam, there have been some reports of adenoviral conjunctivitis [35,36], and gastro-enteritis [12,13,37]. HAdV is among the less common viral causes of seasonal respiratory infections as compared to many other more common viruses such as influenza, parainfluenza, respiratory syncytial virus (RSVs), enteroviruses, and rhinovirus [38–45]. HAdV is among the co-infection factors in patients with severe pneumonia [41]. HAdV-associated pneumonia, especially associated with HAdV-B7 has been reported to be among the causes of death in patients following measles infection [46,47]. There had not been any information regarding HAdV-associated acute respiratory infection (ARI) outbreaks in children in Vietnam until the first half of 2022.

Since the second half of 2022, HAdV had emerged as the pathogen associated with the outbreak of ARI in children in Northern Vietnam. Whilst most of the cases were mild, there had been a considerable number of cases with severe complications such as pneumonia, acute respiratory distress, which required intensive treatment and prolonged hospitalization. In addition, quite a few deaths attributed to HAdV infection had been noted. While the COVID-19 pandemic had been somewhat under control, the outbreak of HAdV- associated ARI in children in Northern Vietnam resulted in a large number of cases and an alarming number of severe cases and deaths which was unprecedented and thus required immediate actions.

Up till now, in Northern Vietnam, detection of HAdV was confirmed by positive PCR of nasal swabs or nasopharyngeal aspirates performed by a few centers. However, molecular typing of HAdV had not been performed to identify the HAdV subtype(s) associated with the outbreak, especially in severe or fatal cases. As such, this study aimed to perform molecular subtyping of HAdV by Sanger sequencing and to examine the clinical contexts of children with HAdV -associated ARI in Northern Vietnam to understand the characteristics of the pathogen and the disease characteristics. The study also examined potential risk factors associated with disease severity to help orientate the treatment and prognosis of the patients.

## Materials & methods

### Ethics statement

The study was approved by the Institutional Ethical Review Board of Vinmec International General Hospital JSC & VinUniversity before being carried out (Reference number: 152/2022/CN-HDDD VMEC). Informed consent for guardian and child was not necessary because clinical data was collected retrospectively from medical records and molecular typing of HAdV was done using nasal swab or nasopharyngeal aspirate samples remaining after PCR assay.

### Sample and data collection

This study was done in Vinmec Times City International Hospital, a private general hospital in Hanoi Vietnam which receives patients mostly from Hanoi and the surrounding provinces. Nasopharyngeal aspirates or nasal swab samples of pediatric inpatients and outpatients with respiratory symptoms suspected of Adenoviral infection were collected in 15.0 ml centrifuge tubes and transported to the Department of Microbiology. Specimens were stored at 4˚C and

subjected to nucleic acid extraction using the QIAamp Viral RNA Kits within 24 hours after collection. Extracted nucleic acids were tested for HAdV and 6 other respiratory pathogens i.e., human enterovirus (HEV), human metapneumovirus (hMPV) and parainfluenza virus 1–4 (PIV1-4) using multiplex real-time PCR Allplex Respiratory Panel 2 (Avicalab Diagnostics, Kenya) according to the manufacturer's instruction [48,49]. Samples with quantification cycle (Ct) value for HAdV < 38 were considered HAdV-positive.

The remaining nasopharyngeal samples (NS) of those tested positive for HAdV were stored and transported to Vinmec Medical Genetic Department for further processing to perform HAdV molecular typing. To ensure a sufficient DNA concentration for sequencing and to minimize possible artifacts due to contamination, only samples with Ct for HAdV <30 were selected for molecular typing based on the 1–6 hypervariable region of HAdV *hexon* gene by Sanger sequencing.

For HAdV-positive pediatric patients, clinical data regarding medical history, especially history of COVID infection, history of hospitalization due to respiratory diseases, clinical manifestations, hematology and biochemistry data as well as treatment and outcome data were retrospectively collected by Vinmec Pediatric Center. Furthermore, microbial assay data which included nasal-pharyngeal swab culture, influenza antigen rapid test, PCR panel for 7 respiratory bacteria, and blood *Mycoplasma pneumoniae* Antibody (IgM) test if available were also retrospectively collected. Study data were collected and managed using REDCap electronic data capture tools hosted at Vinmec Healthcare System [50,51].

All patients included in this study had respiratory symptoms such as cough and runny nose. Patients were classified as having lower respiratory illnesses if they exhibited lower respiratory associated symptoms such as rales, rhonchi, wheezing and/or having lung lesions as indicated by chest x-ray. Patients were classified as having severe pneumonia if they exhibited lower respiratory manifestations as described above, lung lesions as indicated by chest x-ray and respiratory distress requiring oxygen supplementation or respiratory support.

Viral co-detection was defined if a patient was HAdV-positive as indicated by PCR panel and was also positive with at least one other virus in the PCR panel or provided a positive influenza antigen rapid test. All patients included in this study were tested for HAdV and HEV, hMPV and PIV1-4 using the PCR respiratory virus-based panel.

Bacterial co-detection was defined if a patient was HAdV-positive as indicated by PCR panel and was also positive with at least one bacterium with nasopharyngeal fluid culture or with PCR panel of the following respiratory bacteria (*Haemophilus influenzae (HI)*, *Streptococcus pneumoniae (SP)*, *Staphylococcus aureus (SA)*, *Moraxella catarrhalis (MC)*, *Mycoplasma pneumoniae (MP)*, *Chlamydophila pneumoniae (CP)*, *Legionella pneumophila (LP)*) or provided a positive blood *Mycoplasma pneumoniae* antibody (IgM) test.

Any co-detection was defined if a patient had either a viral or bacterial co-detection as described above.

## HAdV molecular typing

The hyper-variable regions 1–6 (HRV$_{1-6}$) of HAdV hexon contain species-specific epitopes, therefore, these sequences were commonly used to identify HAdV subtypes. HAdV molecular typing was performed on 97 HAdV-positive samples. Viral nucleic acids were extracted from NS using QIAamp DNA Blood Mini Kit (QIAGEN, Germany). Following the protocol recommended by CDC, the *hexon* of HAdV was amplified using nested PCR targeting the HVR$_{1-6}$ region which generated amplicons between 764–896 base pairs (bp) [52]. The outer primer pairs used for first round of nested-PCR were AdhexF1 (5′- TICTTTGACATICGIGG

IGTICTIGA-3′) and AdhexR1 (5′- CTGTCIACIGCCTGRTTCCACA-3′). The inner primer pairs used for the second round of nested-PCR were AdhexF2 (5′- GGYCCYAGYT-TYAARCCCTAYTC-3′) and AdhexR2 (5′- GGTTCTGTC ICCCAGAGARTCIAGCA-3′). Nested-PCR was performed using a final concentration of 1x GoTaq Green Master (Mix Promega), 0.4 μM of each primer, 2.0 μl of extracted viral nucleic acid or 0.2 μl of 1st nested-PCR product and made up to a final volume of 25 μl with double distilled water. The PCR thermal program for both rounds of nested-PCR reactions were comprised of an initial denaturation step at 94˚C for 2 min, followed by 11 touched-down cycles of 95˚C for 30 sec, 55˚C (decrease 1˚C after each cycle) for 45 sec, 72˚C for 1 min, followed by 20 thermal cycles of 95˚C for 30 sec, 45˚C for 45 sec, 72˚C for 1 min, and a 5 min final extension at 72˚C. The size of PCR products was checked by DNA electrophoresis analysis using 1% agarose gels stained with RedSafe DNA Stain (20,000 X) (Chembio, USA). Only samples showing amplicons with the expected band size following nested-PCR were used for Sanger sequencing of the $HVR_{1-6}$ of hexon gene.

Each nested-PCR product was sequenced using both forward AdhexF2 and reverse AdhexR2 primers. Sequencing PCR reactions were performed in a final volume of 10 μl comprising 0.5 μl of BigDye Terminator v3.1 Cycle Sequencing (ABI), 2.0 μl of Sequencing Buffer 5X, 0.85 μl (10 μM) of primer, 0.2μl 2nd nested-PCR product, and distilled water (up-to-10 μl). The sequencing thermal cycle for sequencing reactions were comprised of an initial denaturation step at 96˚C for 2 min, followed by 25 thermal cycles of 95˚C for 10 sec, 50˚C for 10 sec, and 60˚C for 2 min. Cycle sequencing products were purified by Bigdye X Terminator purification kit before analysis on the Applied Biosystems 3500 Dx Genetic Analyzer.

## Data analysis

**Sequence data and phylogenetic analysis.**   The sequence data were used to identify the subtype of HAdV using the Basic Local Alignment Search Tool (BLAST) (https://blast.ncbi.nlm.nih.gov/Blast.cgi). The sequences of $HVR_{1-6}$ of *hexon* gene from all samples were aligned with related reference strains obtained from GenBank (reference strains are listed in S1 Table) using CLUSTAL W [53]. The robustness of the phylogenetic tree was constructed by the neighbor-joining method with bootstrap analysis (n = 1000) by iTOL v6 and Adobe Illustrator software. The cut-off value, which refers to the bootstrap support threshold used in constructing the phylogenetic tree, was <80%.

**Analysis of HAdV types and clinical contexts.**   HAdV types and other clinical features were compared between disease contexts such as lower respiratory illnesses vs. no lower respiratory illnesses or severe pneumonia vs no severe pneumonia. The purpose was to examine if HAdV type or any other clinical features were associated with the severity of HAdV infection. Clinical features were also compared between HAdV types to examine if any clinical characteristic was associated with HAdV types.

For comparison between groups for initial association exploration, Mann-Whitney U test or Kruskal-Wallis's test was used for continuous variables and Fisher's exact test was used for categorical variables. All potential risk factor variables for lower respiratory illnesses or severe pneumonia were also explored using univariate logistic regression models. The potential risk factors which were clinically relevant or statistically significant from the above exploration were further examined using multivariate logistic regression models. Unadjusted and adjusted odds ratio (OR) and 95% confidence interval (95%CI) by profile likelihood and p-values from Wald test were reported.

Data analysis was performed using R statistical software version 4.1 [54]. All statistical tests were 2-sided with a significant level alpha of 0.05.

## Results

### HAdV assays and HAdV subtypes

From October 13th to November 9th, 2022, a multiplex RT-PCR based panel for 7 respiratory viruses including HAdV was performed on nasal swabs or nasopharyngeal aspirates of 253 patients with ARI. Using a cycle threshold (Ct) of <38 as an indication of a positive score, samples of 138 patients tested positive for HAdV (with Ct for HAdV <38). Among the HAdV positive samples, 97 patient samples with Ct for HAdV <30 were selected for HAdV molecular subtyping. The median (interquartile range (IQR)) Ct for HAdV of these 97 samples was 23.6 (21.9, 25.6).

Ninety-seven HAdV samples were genotyped successfully. BLAST results of 97 HAdV sequences indicated that 81 (83.5%) samples belonged to subtype B3, 15 (15.5%) samples belonged to subtype B7 and 1 (1.0%) sample belonged to subtype C2. Phylogenetic analysis was applied to compare 97 hexon gene sequences to each other and 24 reference sequences downloaded from GenBank (S1 Table and S1 Data). The phylogenetic tree confirmed the hexon sequences of HAdV samples belonged to three distinguishable clusters of HAdV type i.e., B3, B7 and C2 (Fig 1).

### Patient characteristics and clinical features

Clinical characteristics of the 97 patients who were selected for HAdV molecular subtyping above are summarized in Table 1. About two thirds of the patients resided in Hanoi and one third of the patients resided in surrounding provinces. Twenty-nine percent of the patients reported a history of prior COVID-19 infection, 4% reported a history of allergy and 7% reported a history of prior hospitalization due to respiratory diseases. There were 73 (75%) patients with only upper respiratory symptoms whilst there were 24 (25%) patients who exhibited lower respiratory manifestations of which 5 (5.1%) had severe pneumonia. There were no differences in demographic and previous history of respiratory disease including COVID-19 infection between those with and without lower respiratory manifestations (Table 1).

The mean (95% confidence interval (CI)) age of all patients was 2.7 (2.4, 3) years. Patients who exhibited lower respiratory manifestations had a mean (95%CI) age of 2.3 (1.6, 2.9) years whilst those without lower respiratory manifestation had a mean (95%CI) age of 3.1 (2.6, 3.6) years. Patients with severe pneumonia were the youngest with mean (95%CI) age of 1.8 (0, 3.8)) years. Males accounted for 56% of all patients and there was no gender difference in those with and without lower respiratory illnesses whereas 4/5 (80%) of those with severe pneumonia were males. About 36% of patients were born by Cesarean (C)-section and the percentage was similar between those with and without lower respiratory illnesses while 4/5 (80%) of those with severe pneumonia were born by C-section (p = 0.051). There was no difference in BMI, gestational age, birth weight and duration of breastfeeding between those with and without lower respiratory illnesses or between those with and without severe pneumonia (Table 1).

Day of disease (the duration time (in days) from the date of disease onset to the date of visiting the hospital) of the patients with lower respiratory manifestations was 4.7 days as compared to 3.3 days for patients who did not exhibit lower respiratory manifestations (p = 0.002). Most of the patients had fever (91%) and cough (91%) at disease onset, with 79% exhibiting high fever (>39˚C) and long duration of fever (mean (95%CI) = 6.2 (5.7, 6.8) days). Poor eating (as compared to usual) as reported by caregivers was found more in patients with lower respiratory illnesses than those without. There was no significant difference in symptoms at disease onset such as cough, red eyes, diarrhea, fever maximal temperature, fever duration,

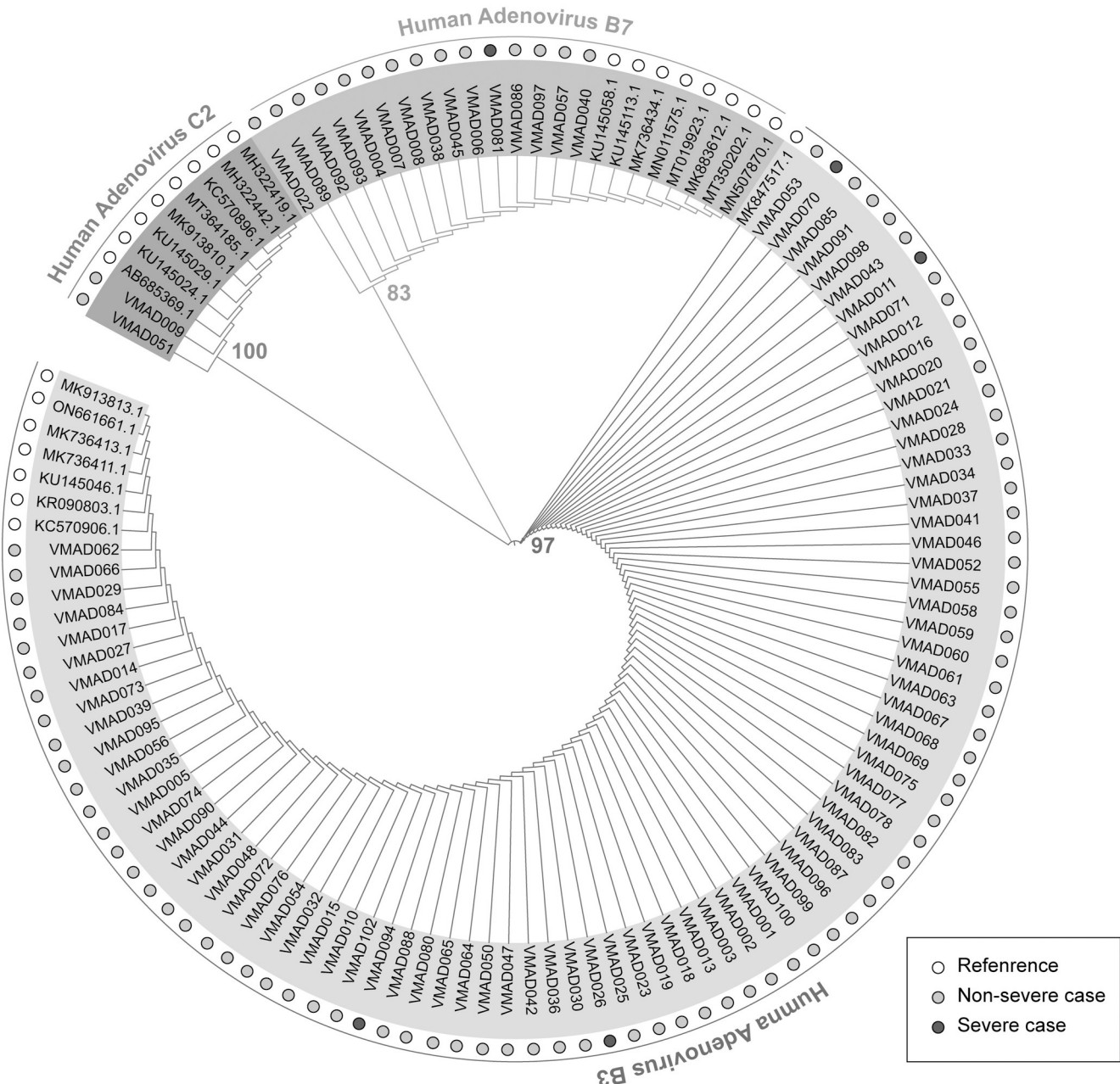

**Fig 1. Human Adenovirus subtypes and phylogenetic tree.** Neighbour-joining phylogenetic of the hexon gene of HAdV strains identified in patients hospitalized with acute respiratory tract infection in Vinmec Times City Hospital between October and November 2022. The branch annotations represent the bootstrap value calculated on 1000 replicates. White, grey and black circles correspond to sequences of HAdV references, HAdV specimens from non-severe patients and HAdV specimens from patients with severe pneumonia.

and ear pain as well as in hematology or biochemistry indexes between those with and without lower respiratory illnesses or between those with and without severe pneumonia. Almost all patients with lower respiratory illnesses (95%) had detectable abnormalities in their chest X-rays as well as 16% of those without lower respiratory symptoms (**Table 1**).

Patients with lower respiratory manifestations received other clinical interventions which included oxygen supplements (25%), IVIG (10%), nebulizers (44%), and general steroids

**Table 1. Characteristics of the patients who underwent Human Adenovirus (HAdV) molecular subtyping.**

| Characteristics | All patients (N = 97) | Group with lower respiratory illnesses (N = 24) | P-value comparison* | Group with severe pneumonia (N = 5) | P-value comparison # |
|---|---|---|---|---|---|
| Sex (males) | 53 (55.8%) | 13 (54.2%) | 1 | 4 (80.0%) | 0.379 |
| Age at hospital visit (year) (mean (95%CI)) | 2.7 (2.4, 3.1) | 2.3 (1.6, 2.9) | 0.141 | 1.8 (0, 3.8) | 0.190 |
| Body Mass Index (BMI) (kg/m2) (mean (95% CI)) | 15. 3 (14.9, 15.6) | 15 (14.2, 15.9) | 0.497 | 14.7 (13.7, 15.7) | 0.487 |
| Gestational age (week) (mean (95%CI)) | 37.6 (36.8, 38.5) | 37.8 (36.9, 38.7) | 0.517 | 36 (36, 36) | 0.115 |
| Delivery method | | | 0.116 | | 0.051 |
| - Cesarean-section | 27 (35.5%) | 11 (50.0%) | | 4 (80.0%) | |
| - Vaginal | 49 (64.5%) | 11 (50.0%) | | 1 (20.0%) | |
| Breast feeding duration (month) (mean (95% CI)) | 12.1 (10.9, 13.3) | 13.2 (10.8, 15.6) | 0.410 | 12.3 (2.7, 21.8) | 0.575 |
| History of COVID-19 infection | 18 (29.0%) | 4 (26.7%) | 1 | 1 (50.0%) | 0.500 |
| History of allergy | 4 (4.3%) | 1 (4.2%) | 1 | 0 (0.0%) | 1 |
| History of immuno-compromise diseases | 0 | 0 | NA | 0 | NA |
| History of chronic disease | 0 | 0 | NA | 0 | NA |
| History of hospitalization due to respiratory disease | 7 (7.2%) | 1 (4.2%) | 0.677 | 1 (20.0%) | 0.318 |
| Day of disease at hospital visit (mean (95%CI)) | 3.6 (3.2, 4) | 4.7 (3.7, 5.6) | 0.002 | 4 (2, 6) | 0.422 |
| Cough | 88 (90.7%) | 23 (95.8%) | 1 | 4 (80.0%) | 0.203 |
| Fever | 88 (90.7%) | 21 (87.5%) | 0.685 | 5 (100.0%) | 1.000 |
| Duration of fever (day) (mean (95%CI)) | 6.2 (5.7, 6.8) | 7.1 (5.7, 8.6) | 0.125 | 6.2 (1.8, 10.6) | 0.839 |
| Red eye | 30 (33.3%) | 5 (21.7%) | 0.207 | 0 (0.0%) | 0.165 |
| Loose stool | 15 (16.7%) | 6 (25.0%) | 0.352 | 1 (20.0%) | 1 |
| Lymph node enlargement | 6 (6.7%) | 1 (4.2%) | 1 | 0 (0.0%) | 1 |
| Skin rash | 1 (1.1%) | 0 (0.0%) | 1 | 0 (0.0%) | 1 |
| Coughing up sputum | 86 (93.5%) | 22 (91.7%) | 1 | 5 (100.0%) | 1 |
| Ear pain | 5 (5.9%) | 1 (4.3%) | 0.653 | 0 (0.0%) | 1 |
| Abdominal pain | 7 (7.9%) | 2 (8.7%) | 1 | 0 (0.0%) | 1 |
| Poor eating | 38 (42.2%) | 16 (66.7%) | 0.007 | 4 (80.0%) | 0.158 |
| White blood cell (WBC) count (x10^9/L) (mean (95%CI)) | 15 (13.5, 16.5) | 13.8 (10.9, 16.7) | 0.295 | 13.5 (3.6, 23.5) | 0.661 |
| Neutrophil count (x10^9/L) (mean (95%CI)) | 9.5 (8.3, 10.8) | 8.8 (6.2, 11.4) | 0.659 | 9.6 (1.1, 18.1) | 0.496 |
| Lymphocyte count (x10^9/L) (mean (95%CI)) | 3.6 (3.2, 4.0) | 3.4 (2.7, 4.1) | 0.405 | 2.8 (1.0, 4.5) | 0.766 |
| Hemoglobin (g/L) (mean (95%CI)) | 85.5 (74.7, 96.3) | 84.6 (64.2, 105) | 0.364 | 89.1 (34.5, 143.7) | 0.538 |
| Platelet (x10^9/L) (mean (95%CI)) | 295 (275, 315) | 304 (250, 357) | 0.811 | 412 (183, 641) | 0.104 |
| C Reactive Protein (CRP) (mg/dL) (mean (95%CI)) | 40.4 (32.8, 48.1) | 41.3 (22.3, 60.3) | 0.773 | 37.2 (0.5, 73.9) | 0.949 |
| Chest X-ray abnormality | 22 (51.2%) | 19 (95.0%) | < 0.001 | 5 (100.0%) | 0.048 |
| Duration of treatment (day) (mean (95%CI)) | 5 (4.5, 5.6) | 6.1 (4.7, 7.4) | 0.020 | 9.5 (4.9, 14.1) | 0.003 |
| Oxy supplement | 6 (6.2%) | 6 (25.0%) | <0.001 | 4 (80.0%) | < 0.001 |
| Intravenous Immunoglobulin (IVIG) infusion for Adenovirus treatment | 2 (2.9%) | 2 (9.5%) | 0.098 | 2 (40.0%) | 0.004 |
| Nebulizer | 10 (14.3%) | 10 (43.5%) | < 0.001 | 4 (80.0%) | 0.001 |
| Corticoid use | 5 (7.4%) | 5 (23.8%) | 0.002 | 4 (80.0%) | < 0.001 |
| Antibiotic use | 78 (86.7%) | 23 (95.8%) | 0.170 | 5 (100.0%) | 1.000 |
| Clinical outcome | | | 0.6941 | | 0.0630 |
| - Dead | 0 (0.0%) | 0 (0.0%) | | 0 (0.0%) | |

*(Continued)*

**Table 1.** (Continued)

| Characteristics | All patients (N = 97) | Group with lower respiratory illnesses (N = 24) | P-value comparison* | Group with severe pneumonia (N = 5) | P-value comparison # |
|---|---|---|---|---|---|
| - Discharged with complication | 1 (1.2%) | 1 (4.8%) | | 1 (20.0%) | |
| - Discharged without complication | 82 (98.8%) | 20 (95.2%) | | 4 (80.0%) | |

*Comparison between those with and without lower respiratory illnesses.

#Comparison between those with and without severe pneumonia.

P-values from Mann-Whitney's test for continuous variables or Fisher's exact test for categorical variables.

The calculated percentage might vary for different variables because missing values varied for different variables (the denominators might vary for different variables due to missing values).

(24%) while patients without lower respiratory manifestations did not receive any of these interventions. The duration of hospitalization was longer for those with lower respiratory manifestations (mean (95%CI) = 6.1 (4.7, 7.4) days) than those without (mean (95%CI) = 4.5 (3.8, 5.1) days) and longest in those with severe pneumonia (mean (95%CI) = 9.5 (4.9, 14.1) days). All patients except one were discharged without complication (**Table 1**).

The five patients with severe pneumonia were of a younger age, predominantly male (80%), predominantly born by C-section (80%) ($p<0.05$ as compared to other patients). There was no other remarkable difference in other demographic, medical history or clinical characteristics when compared to other patients (**Table 1**).

## Other microbiology assays and co-detection of other respiratory viruses or bacteria

The data regarding other microbiology assays performed for the 97 patients who underwent HAdV molecular subtyping are summarized in **Table 2**. Sixty-seven (69.1%) patients underwent influenza rapid antigen test of which 18 patients exhibited lower respiratory manifestations and 49 patients did not have lower respiratory manifestations. Bacterial cultures of nasopharyngeal aspirates were performed for 58 (59.8%) patients which included 21 patients with lower respiratory manifestations and 37 patients without lower respiratory manifestations. Mycoplasma antibody (IgM) blood test and PCR panel for 7 respiratory bacteria were performed with 2 (2.1%) and 1 (1%) of all included patients, respectively (**Table 2**).

Regarding co-detection of other viruses, in addition to HAdV, 19 (approx. 20%) patients tested positive for at least one other virus from the respiratory virus PCR panel of which 5 (5.1%) patients were co-detected with at least 2 other viruses. The number of samples co-detected with HEV, MPV, PIV1, PIV2, PIV3 and PIV4 in the PCR panel were 7, 9, 4, 2, 0, and 2 respectively. There was one patient co-detected with Influenza virus by rapid antigen test. In overall, there were 20 patients co-detected with at least 1 virus by either PCR panel or Influenza rapid antigen test (**Table 3**).

With respect to co-detection of respiratory bacteria, 3, 11, and 5 patients tested positive for Streptococcus pneumoniae, Haemophilus influenzae, and Moraxella catarrhalis respectively by culture of nasopharyngeal aspirates. There were 2 patients whose blood antibody test (IgM) was positive for Mycoplasma pneumoniae. Overall, 20 patients co-detected for at least one bacterium by either nasopharyngeal fluid culture or blood antibody test. Finally, 5 patients co-detected for at least one virus and at least one bacterium. In total, there were 35 patients co-detected with at least one either virus or bacteria (**Table 3**).

The percentage of samples co-detected with at least one other respiratory virus was significantly higher in patients with lower respiratory manifestations compared to those without (13/

**Table 2. Microbiology characteristics of the patients who underwent Human Adenovirus (HAdV) molecular subtyping.**

| Microbiology characteristics | All patients (N = 97) | Group with lower respiratory illnesses (N = 24) | P-value comparison* | Group with severe pneumonia (N = 5) | P-value comparison # |
|---|---|---|---|---|---|
| Performed PCR panel for 7 respiratory viruses$ | 97 (100%) | 24 (100%) | NA | 5 (100%) | NA |
| Performed Influenza rapid antigen test | 67 (69.1%) | 18 (75.0%) | 0.269 | 2 (40.0%) | 1 |
| Performed culture of nasopharyngeal aspirates | 58 (59.8%) | 21 (87.5%) | 1 | 5 (100%) | 0.318 |
| Performed Mycoplasma antibody (IgM) blood test | 2 (2.1%) | 1 (4.2%) | 0.436 | 2 (40.0%) | 1 |
| Performed PCR panel for 7 respiratory bacteria@ | 1 (1.0%) | 0 | NA | 0 | NA |
| HAdV type | | | 0.636 | | 0.602 |
| - Type B3 | 81(83.5%) | 19 (79.2%) | | 4 (80.0%) | |
| - Type B7 | 15 (15.5%) | 5 (20.8%) | | 1 (20.0%) | |
| - Type C2 | 1 (1.0%) | 0 | | 0 | |
| Viral co-detection$ $ | 20 (20.6%) | 13 (54.2%) | < 0.001 | 4 (80.0%) | 0.006 |
| Bacterial co-detection @@ | 20 (20.6%) | 7 (29.2%) | 0.253 | 3 (60.0%) | 0.058 |
| Any co-detection $@ | 35 (36.1%) | 17 (70.8%) | < 0.001 | 5 (100.0%) | 0.005 |

*Comparison between those with and without lower respiratory illnesses.

#Comparison between those with and without severe pneumonia.

P-values from Fisher's exact test for categorical variables.

$PCR panel for 7 respiratory viruses (human adenovirus (HAdV), human enterovirus (HEV), human metapneumovirus (hMPV) and parainfluenza virus 1–4 (PIV1-4)) was performed for all patients included in this study.

@PCR panel of 7 respiratory bacteria (*Haemophilus influenzae (HI)*, *Streptococcus pneumoniae (SP)*, *Staphylococcus aureus (SA)*, *Moraxella catarrhalis (MC)*, *Mycoplasma pneumoniae (MP)*, *Chlamydophila pneumoniae (CP)*, *Legionella pneumophila (LP)*).

$ $Co-detection with ≥1 other viruses in PCR panel or influenza virus type A or type B using rapid antigen test.

@@Co-detection with ≥1 bacteria in nasopharyngeal fluid culture or PCR panel of 7 respiratory bacteria (*Haemophilus influenzae (HI)*, *Streptococcus pneumoniae (SP)*, *Staphylococcus aureus (SA)*, *Moraxella catarrhalis (MC)*, *Mycoplasma pneumoniae (MP)*, *Chlamydophila pneumoniae (CP)*, *Legionella pneumophila (LP)*) or *Mycoplasma pneumoniae antibody (IgM) test*.

$@Any co-detection of ≥1 bacteria or viruses mentioned above.

24(54%) vs. 7/73 (9.6%), p< 0.001) and especially higher in those with severe pneumonia vs. those without (4/5 (80%) vs. 16/92 (17.4%), p = 0.006). Co-detection with at least one respiratory bacterium was detected in 20 (20.6%) patients, with 7 of these patients (29.2%) experiencing lower respiratory illnesses (p = 0.253) and in 3 (60.0%) with severe pneumonia (p = 0.058). Any co-detection of respiratory viruses or bacteria was found in 35 (36.1%) of all patients, significantly higher in those with lower respiratory illnesses (17 (70.8%), p< 0.001) and especially higher in those with severe pneumonia (5 (100.0%), p = 0.005) (**Table 2**).

### Risk factors for lower respiratory illnesses and severe pneumonia

Potential risk factors for lower respiratory illnesses or severe pneumonia which were clinically relevant and statistically significant from the above exploration included the following: (1) viral co-detection of HAdV with ≥1 respiratory viruses or bacterial co-detection of HAdV with ≥1 respiratory bacteria or any co-detection of HAdV with ≥1 respiratory viruses or bacteria, (2) age of the patient at hospital visit, (3) sex, (4) birth delivery method, and (5) day of disease at hospital visit. The results from univariate and multivariate logistic regression analysis adjusting for these factors in the models for the comparison between those with lower

**Table 3. Viral and bacterial co-detection with Human Adenovirus (HAdV).**

| Viral co-detection | Any virus | HEV* | MPV* | PIV1* | PIV2* | PIV3* | PIV*4 | Influenza# |
|---|---|---|---|---|---|---|---|---|
| Number of patients | 20 | 7 | 9 | 4 | 2 | 0 | 2 | 1 |
| **Bacterial co-detection** | **Any bacteria** | **Streptococcus pneumoniae $** | **Haemophilus influenzae $** | **Moraxella catarrhalis $** | **Mycoplasma pneumoniae @** | **Others** | | |
| Number of patients | 20 | 3 | 11 | 5 | 2 | 0 | | |
| **Viral and bacterial co-detection** | **Any virus and bacteria** | | | | | | | |
| Number of patients | 5 | | | | | | | |
| **Either viral or bacterial co-detection** | **Any virus or bacteria** | | | | | | | |
| Number of patients | 35 | | | | | | | |

*Viruses in the same PCR panel with human Adenovirus for nasal swab or nasopharyngeal aspirates: HEV: human enterovirus, hMPV: human metapneumovirus,

PIV1-4: parainfluenza virus 1–4.

#By rapid antigen test for nasal swabs

$By culture of nasopharyngeal fluid

@By blood antibody test (IgM)

respiratory illnesses or severe pneumonia vs. those without are summarized in **Table 4**. Co-detection of HAdV with ≥1 other respiratory viruses was significantly associated with either lower respiratory illnesses or severe pneumonia in univariate analysis and also after adjusting for age at hospital visit, sex, birth delivery method, day of disease at hospital visit. The adjusted OR (95%CI); (p-value) of viral co-detection for those with lower respiratory manifestations compared to those without lower respiratory manifestations was 10.74 (2.83, 48.17); 0.0009 and for those with severe pneumonia compared to those without severe pneumonia was 19.44 (2.12, 492.73); 0.0207 respectively. Co-detection of HAdV with any respiratory viruses or bacteria was also remarkably significantly associated with lower respiratory manifestations (adjusted OR (95%CI); (p-value) = 5.21 (1.60, 19.36); 0.0084 compared to those without lower respiratory manifestations).

## The relationship between HAdV types and clinical features

Those infected with HAdV type B3 had higher white blood cell (WBC) count (mean (95%CI) = 16.2 (14.6, 17.8) x10^9/L) and higher neutrophil count (mean (95%CI) = 10.7 (9.3, 12.1) x10^9/L) as compared to those detected with HAdV type B7 (mean (95%CI) of WBC and neutrophil count were 9.4 (7.1, 11.7) and 4.4 (3.0, 5.8) x10^9/L, respectively) (p<0.001). In general, there was no remarkable difference in demographic, medical history, disease manifestation or outcome features between patients detected with HAdV type B3, B7, and C2 in this study except the above mentioned hematology test (**Fig 2** and **S2 Table**).

## Discussion

This study, to the best of our knowledge is probably the first providing a quick comprehensive snapshot of both clinical features and molecular typing of the HAdV-associated ARI outbreak in children in northern Vietnam at the end of 2022.

The predominant HAdV type in this study was B3, followed by B7 and C2. These HAdV subtypes are similar to the HAdV subtypes reported by some other studies on circulating or outbreaks of respiratory HAdV infections in children before the COVID-19 pandemic in Asia such as studies on circulating respiratory HAdV in Hong Kong in 2014 [55], Taiwan from 2002 to 2013 [8], Guangzhou China from 2017 to 2019 [5] or a study on febrile respiratory

**Table 4. Univariate and multivariate logistic regression analysis of risk factors of lower respiratory illnesses or severe pneumonia.**

| | Lower respiratory illnesses | | Severe pneumonia | |
|---|---|---|---|---|
| Features | Univariate OR (95%CI); p-value* | Multivariate OR (95%CI); p-value ** | Univariate OR (95%CI); p-value# | Multivariate OR (95%CI); p-value## |
| Viral co-detection$ (yes vs. no) | 11.14 (3.77, 36.05); <0.0001 | 10.74 (2.83, 48.17); 0.0009 | 19.00 (2.60, 385.24); 0.0105 | 19.44 (2.12, 492.73); 0.0207 |
| Bacterial co-detection (yes vs. no)@ | 1.90 (0.63, 5.44); 0.2373 | 1.04 (0.26, 3.83); 0.9506 | 6.62 (1.02, 53.28); 0.0470 | 19.25 (1.74, 528.71); 0.0297 |
| Any co-detection (yes vs. no)$@ | 7.42 (2.75, 21.98); 0.0001 | 5.21 (1.60, 19.36); 0.0084 | Inf (NA,NA); NA ### | Inf (NA,NA); NA ### |
| Age (year) | 0.71 (0.48, 1.00); 0.0678 | 0.72 (0.41, 1.18); 0.2175 | 0.56 (0.24, 1.13); 0.1462 | 0.86 (0.32, 2.21); 0.7466 |
| Sex (male vs. female) | 0.92 (0.36, 2.35); 0.8531 | 0.52 (0.13, 1.86); 0.3226 | 3.35 (0.47, 66.86); 0.2884 | 1.46 (0.11, 37.77); 0.7823 |
| Delivery method (vaginal vs. C-section) | 0.42 (0.15, 1.17); 0.0964 | 0.45 (0.12, 1.63); 0.2235 | 0.12 (0.01, 0.87); 0.0642 | 0.11 (0.00, 1.14); 0.1002 |
| Day of disease at hospital visit& | 1.46 (1.13, 1.95); 0.0060 | 1.48 (1.09, 2.13); 0.0204 | 1.10 (0.66, 1.63); 0.6483 | 1.21 (0.65, 2.10); 0.4884 |

*Comparison between those with and without lower respiratory illnesses. 95% confidence interval (CI) from profile likelihood and p-values from Wald test of univariate logistic regression model against each variable.

**Comparison between those with and without lower respiratory illnesses. 95%CI from profile likelihood and p-values from Wald test for each variable in the multivariate logistic regression model. For viral co-detection or bacterial co-detection or any co-detection, estimates were from the multivariate logistic regression model containing either viral co-detection or bacterial co-detection or any co-detection and 4 other variables. For 4 other variables (age, sex, delivery method, day of disease at hospital visit), estimates were from the multivariate logistic regression model containing viral co-detection and these 4 variables.

# Comparison between those with and without severe pneumonia. 95%CI from profile likelihood and p-values from Wald test of univariate logistic regression model against each variable.

## Comparison between those with and without severe pneumonia. 95%CI from profile likelihood and p-values from Wald test for each variable in the multivariate logistic regression model. For viral co-detection or bacterial co-detection or any co-detection, estimates were from the multivariate logistic regression model containing either viral co-detection or bacterial co-detection or any co-detection and 4 other variables. For 4 other variables (age, sex, delivery method, day of disease at hospital visit), estimates were from the multivariate logistic regression model containing viral co-detection and these 4 variables.

###Estimates from logistic models were not computed because any co-detection was found in all (100%) cases with severe pneumonia.

$Viral co-detection with ≥1 other viruses in the PCR panel (HEV: human enterovirus, hMPV: human metapneumovirus, PIV1-4: parainfluenza virus 1–4) or influenza virus type A or type B using rapid antigen test

@Bacterial co-detection with ≥1 bacteria via nasopharyngeal fluid culture or PCR panel of 7 respiratory bacteria (*Haemophilus influenzae (HI), Streptococcus pneumoniae (SP), Staphylococcus aureus (SA), Moraxella catarrhalis (MC), Mycoplasma pneumoniae (MP), Chlamydophila pneumoniae (CP), Legionella pneumophila (LP)*) or *Mycoplasma pneumoniae antibody (IgM) test.*

$@Any co-detection of ≥1 bacteria or viruses mentioned above.

&Duration (in day) from the date of disease onset to the date of hospital visit.

HAdV outbreaks in Hangzhou China in 2011 [10] or a study on successive respiratory HAdV outbreaks in Seoul, Korea from 1990 to 2000 [6]. Novel genome types were identified during the outbreaks of lower respiratory tract HAdV infection in Seoul Korea and this 11-year study also suggested a genome type shift between successive outbreaks [6]. Therefore, further studies examining the genome types or variants of HAdV would be useful.

There was no significant association between the 3 HAdV subtypes identified and clinical features in this study, except that WBC and neutrophil numbers were higher in those detected with HAdV type B3 than with type B7. The proportion of severe cases seemed to have similar distribution between the more predominant HAdV type B3 and type B7. This is consistent with other previously published studies in which HAdV type B3 and B7 were the most common types among more severe respiratory HAdV infections [6,8,10,26,27,29–31]. These findings suggest that HAdV subtypes may not be associated with the severity of ARI in children in our study.

Co-detections of HAdV with other respiratory viruses stood out to be the most significant factor associated with the increased severity of respiratory diseases. This association remained statistically significant in a multivariate analysis adjusting for other potential risk factors which

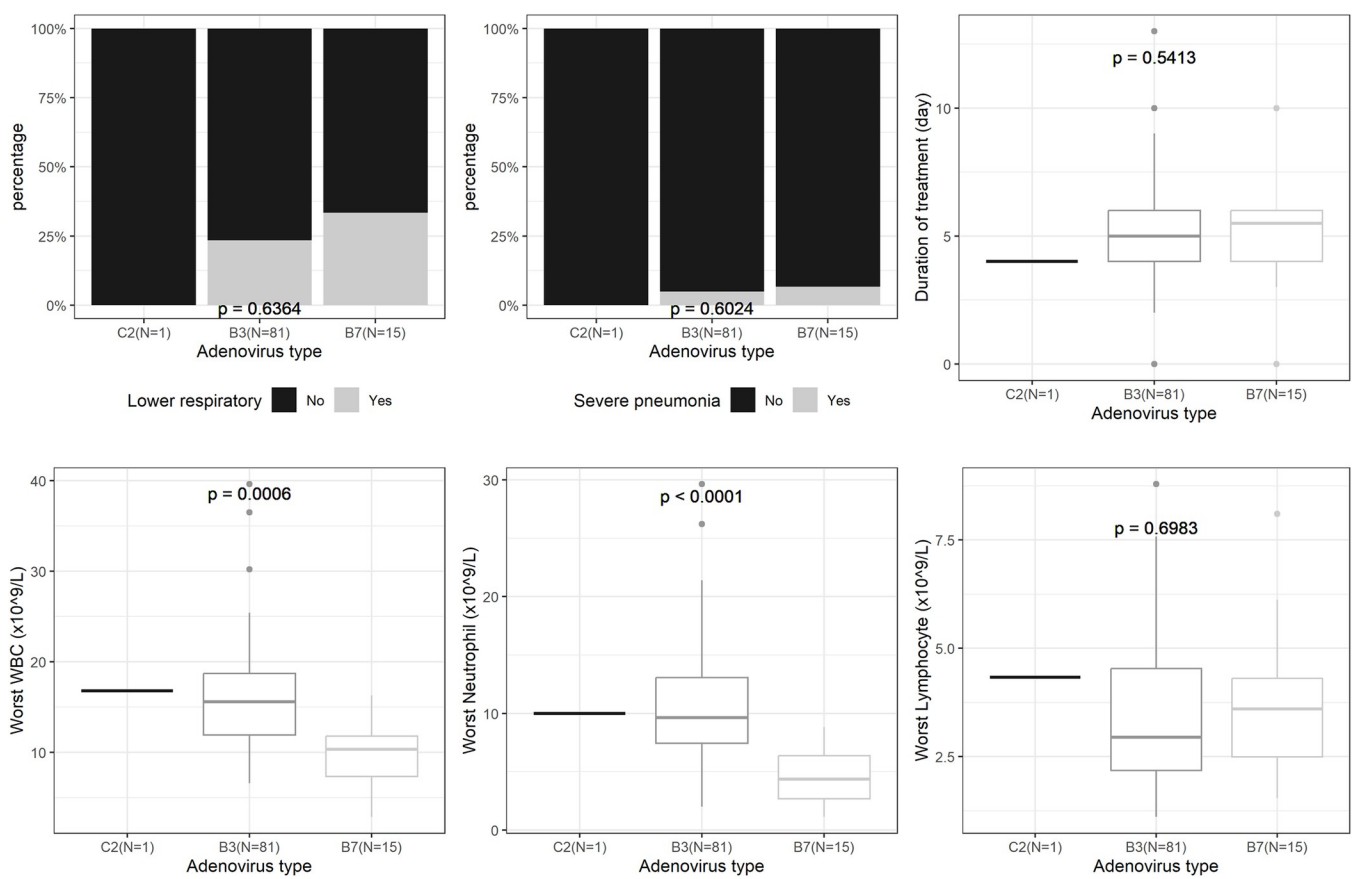

**Fig 2. Patients clinical features by human adenovirus types.** Upper panel: some clinical features (lower respiratory illnesses (yes/ no), severe pneumonia (yes/ no), duration of treatment in days) of three groups of patients detected with human Adenovirus (HAdV) type C2 vs. B3 vs. B7. Lower panel: some hematological features (worst white blood cell count (WBC) (x10^9/L), worst Neutrophil count (x10^9/L), worst Lymphocyte count (x10^9/L)) of three groups of patients detected with human Adenovirus (HAdV) type C2 vs. B3 vs. B7.

were either clinically relevant or statistically associated with lower respiratory illnesses or severe pneumonia. Co-detection of other viruses (HEV, MPV, and PIV 1–4) together with HAdV in the same PCR panel was the main driver (19/20 cases with viral co-detection) for this significant association. This PCR panel was performed for all patients included in this report who had either upper or lower, mild or severe respiratory manifestations. Therefore, the association is both clinically meaningful and statistically robust. This finding is also consistent with a large study in the US which reported that HAdV co-detection with other respiratory viruses was associated with greater disease severity among children with ARI compared to HAdV detection alone [56]. This remarkable association suggests that HAdV might be among the causes of the ARI outbreak in children but might not be the main cause of severe cases. Instead, co-detection of other respiratory viruses together with HAdV might increase the severity of the disease. This advocates the advantage of multi-factor respiratory PCR panel(s) for common respiratory viruses and probably also respiratory bacteria together with other microbial tests such as culture for diagnosis and prognosis of children with ARI. These results may be helpful for the diagnosis, treatment, and prognosis of patients in future outbreaks of HAdV-associated ARI in children.

A general limitation of a study like this one is that it is not possible to clearly differentiate the clinical symptoms of HAdV and of other respiratory pathogens in pediatric patients with ARI. As

stated in some published reports, it is difficult to differentiate between colonization and infection or between co-infection and co-detection for the detected respiratory microorganisms [57]. Blood serologic tests were not performed for the viruses or bacteria detected from nasal swabs or nasopharyngeal aspirates in this study. As such, there may not be enough evidence to claim that the ARI outbreak was caused by HAdV infection and thus for this study it would be more reasonable to use "HAdV-associated ARI outbreak" instead. Another limitation of this study is that Respiratory syncytial virus (RSV), a common ARI pathogen in young children, was not part of the PCR panel for respiratory viruses together with HAdV and was not tested. We used commercial kit Allplex Respiratory Panel 2 to detect HAdV and the other 6 ARI viruses which unfortunately did not include RSV. Therefore, some respiratory viruses including RSV were left out in this study due to this technical limitation. Besides, our study sample size was relatively small but was similar to those of many other previously published studies analysing molecular typing for HAdV outbreaks [21,25,29,55,58,59]. Next, the study samples were from pediatric patients of a private general hospital in Hanoi, Vietnam and molecular typing was performed using samples with Ct for HAdV <30. As such, the study samples may not fully represent the general pediatric population of the ARI outbreak. Additionally, it should be noted that clinical data were collected retrospectively for samples with HAdV-positive PCR and thus some clinical data features may be incomplete, especially for outpatients. Moreover, typing of HAdV based on $HRV_{1-6}$ region of HadV hexon may be reasonable for rapid identification of HAdV types during an outbreak as used in this study or for routine large-scale molecular epidemiological monitoring. However, typing of HAdV based on $HRV_{1-6}$ region of HadV hexon does not provide enough molecular typing data to identify the new variants across the pathogen genomes or recognize intertypic recombination. In a previous study, HAdVs has been reported to recombine for the emergence of new strains. Whole genome sequencing (WES) analysis demonstrated that the new HAdV species C had the highest homology to HAdV-C89 but possessed the fiber gene of HAdV-C1 [60]. Therefore, further research to examine HAdV genome types or variants to investigate others possible alternations or modifications of nucleotide on whole HAdV genomics together with more comprehensive evaluation of the patient characteristics is recommended.

In summary, this study revealed that HAdV type B3 and type B7 were predominant in the outbreak of HAdV-associated ARI in children in northern Vietnam and HAdV type might not be associated with the severity of diseases. Instead, co-detection of HAdV together with other respiratory pathogens, especially viruses appeared to be a significant risk factor for lower respiratory tract illnesses and severe pneumonia. These findings advocate the advantages of multifactor microbial panels for the diagnosis and prognosis of respiratory infections in children. These results may be helpful for the diagnosis, treatment, and prognosis of patients in future outbreaks of HAdV-associated ARI in children.

## Supporting information

**S1 Table. List of reference Human Adenovirus strains used in the manuscript's *hexon* gene phylogenetic comparisons.**
(DOCX)

**S2 Table. Microbiology and patient characteristics by Human Adenovirus subtypes.**
(DOCX)

**S1 Data. Nucleotide sequence of reference Human Adenovirus strains and sequencing results.**
(XLSX)

**S2 Data. Patients clinical data and corresponding adenovirus subtypes.**
(XLSX)

**S1 Protocol. Molecular typing protocol of Adenovirus by Sanger Sequencing.**
(PDF)

## Author Contributions

**Conceptualization:** Dinh-Dung Nguyen, Huyen Thi Thanh Tran, Ha Thi Thanh Ly, Nhan Thi Ho.

**Data curation:** Dinh-Dung Nguyen, Lan Tuyet Phung, Nhan Thi Ho.

**Formal analysis:** Dinh-Dung Nguyen, Nhan Thi Ho.

**Funding acquisition:** Dinh-Dung Nguyen, Huyen Thi Thanh Tran, Ha Thi Thanh Ly.

**Investigation:** Dinh-Dung Nguyen, Lan Tuyet Phung, Nhung Phuong Dinh, Phuong Mai Doan, Anh Thi Nguyen, Luc Danh Dang, Thia Thi Doan, Khuong Thi Pham, Huong Lan Pham, Dai Xuan Hoang, Thao Ngoc Pham, Bao Thai Tran, Trang Thi Thuc Tran, Nhan Thi Ho.

**Methodology:** Nhan Thi Ho.

**Software:** Nhan Thi Ho.

**Supervision:** Huyen Thi Thanh Tran, Ha Thi Thanh Ly, Huong Thi Minh Le, An Nhat Pham, Antony Antoniou, Nhan Thi Ho.

**Validation:** Nhan Thi Ho.

**Visualization:** Anh Hang Mai Vo, Nhan Thi Ho.

**Writing – original draft:** Dinh-Dung Nguyen, Nhan Thi Ho.

**Writing – review & editing:** Lan Tuyet Phung, Huyen Thi Thanh Tran, Ha Thi Thanh Ly, Anh Hang Mai Vo, Nhung Phuong Dinh, Phuong Mai Doan, Anh Thi Nguyen, Luc Danh Dang, Thia Thi Doan, Khuong Thi Pham, Huong Lan Pham, Dai Xuan Hoang, Thao Ngoc Pham, Bao Thai Tran, Trang Thi Thuc Tran, Huong Thi Minh Le, An Nhat Pham, Antony Antoniou, Nhan Thi Ho.

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
