## [Decision Letter · Decision Letter 0]

19 Jun 2023

Dear Dr. Nguyen,

Thank you very much for submitting your manuscript "Molecular subtypes of respiratory Adenovirus infection outbreak in children in Northern Vietnam and risk factors of more severe cases" for consideration at PLOS Neglected Tropical Diseases. As with all papers reviewed by the journal, your manuscript was reviewed by members of the editorial board and by several independent reviewers. In light of the reviews (below this email), we would like to invite the resubmission of a significantly-revised version that takes into account the reviewers' comments. 

We cannot make any decision about publication until we have seen the revised manuscript and your response to the reviewers' comments. Your revised manuscript is also likely to be sent to reviewers for further evaluation.

Sincerely,

Chukwunonso Nzelu, Ph.D.

Guest Editor

Elvina Viennet

Section Editor

Reviewer's Responses to Questions

**Key Review Criteria Required for Acceptance?**

**Methods**

-Are the objectives of the study clearly articulated with a clear testable hypothesis stated?

-Is the study design appropriate to address the stated objectives?

-Is the population clearly described and appropriate for the hypothesis being tested?

-Is the sample size sufficient to ensure adequate power to address the hypothesis being tested?

-Were correct statistical analysis used to support conclusions?

-Are there concerns about ethical or regulatory requirements being met?

Reviewer #1: (No Response)

Reviewer #2: -Are the objectives of the study clearly articulated with a clear testable hypothesis stated? Yes

-Is the study design appropriate to address the stated objectives? No

-Is the population clearly described and appropriate for the hypothesis being tested? No

-Is the sample size sufficient to ensure adequate power to address the hypothesis being tested? Yes

-Were correct statistical analysis used to support conclusions? Yes

-Are there concerns about ethical or regulatory requirements being met? No

Reviewer #3: (No Response)

Reviewer #4: - Was the presence of RSV in the clinical specimens investigated? Is there a reason why this prevalent respiratory pathogen was not included?

-Was fever part of the diagnostic criteria?

-The acronym for human adenovirus is HAdV. Please correct throughout the manuscript. 

-On line 124 the correct term to describe the coding region targeted for amplification and molecular typing is HEXON, not exon.

-What is the limit of detection of the diagnostic PCR used to detect HAdV in the study?

-On line 152 please provide the URL for NCBI.

-The spelling of GenBank needs to be corrected.

-On line 154 a reference needs to be cited for ClustalW.

-What parameter is the cutoff value of <80% referring to?

-On line 173, please correct the text to read qPCR.

**Results**

-Does the analysis presented match the analysis plan?

-Are the results clearly and completely presented?

-Are the figures (Tables, Images) of sufficient quality for clarity?

Reviewer #1: (No Response)

Reviewer #2: -Does the analysis presented match the analysis plan? Yes

-Are the results clearly and completely presented? No

-Are the figures (Tables, Images) of sufficient quality for clarity? No

Reviewer #3: The analysis presented matches the analysis plan and the results are clearly presented as well as discussed in the context of the aim. The numbers stated in Table 1 for the HAdV-3 and 7 does not match what is stated in the discussion. This could just be a minor typing or calculation error

Reviewer #4: -Was a documented previous COVID-19 infection a risk factor for HAdV-associated disease severity?

-On line 226, what does "day of illness" mean?

-Please use the symbol "°C to describe temperature.

-On line 311, please correct the virus type to read B7.

-Were the hexon amplicon sequences deposited in GenBank?

-The legend of figure 3 needs revision.

**Conclusions**

-Are the conclusions supported by the data presented?

-Are the limitations of analysis clearly described?

-Do the authors discuss how these data can be helpful to advance our understanding of the topic under study?

-Is public health relevance addressed?

Reviewer #1: (No Response)

Reviewer #2: -Are the conclusions supported by the data presented? No

-Are the limitations of analysis clearly described? No

-Do the authors discuss how these data can be helpful to advance our understanding of the topic under study? Yes 

-Is public health relevance addressed? Yes

Reviewer #3: The conclusions are supported by the data presented and limitation of the analysis are clearly described but recommendation for further research are not discussed. Discussion on how the data can be helpful to advance our understanding on adenovirus infections and associated risks for severe disease are clearly stated. It is clear to the reader that HAdV3 and 7 were the predominant types in the outbreak of respiratory HAdV infection among children in Northern Vietnam and that co-infection of HAdV with other respiratory pathogens is a significant risk factor for severe respiratory infection

Reviewer #4: This study provides molecular epidemiology data for HAdV respiratory infections in Vietnam for the first time since the early 2000. The study provides important-albeit modest- information regarding prevalent HAdV types circulating in the area.

**Editorial and Data Presentation Modifications?**

Reviewer #1: (No Response)

Reviewer #2: (No Response)

Reviewer #3: (No Response)

Reviewer #4: The wording of the title of this manuscript needs revision in many sections.

**Summary and General Comments**

Reviewer #1: General comments:

Human adenoviruses (HAdVs) are a group of viruses that can cause various illnesses in humans, including respiratory, gastrointestinal, and ocular infections. There are over 50 known serotypes of adenoviruses that can cause human infections. Some serotypes, such as Adenovirus serotypes 3, 4, and 7, are known to be more commonly associated with respiratory tract infections in children. Different serotypes can have varying levels of pathogenicity and clinical manifestations. 

This study aimed to investigate the molecular types of HAdVs outbreak and attempted to determine their associations with clinical manifestations of children with respiratory infections in Northern Vietnam in 2022. 

Several flaws as outlined below:

1. Abstract. Methods. It is suggested to described that all the enrolled patients were presented with acute lower respiratory tract infections. Besides, according to the text, the samples used for sequencing included nasopharyngeal aspirates and nasal swabs. 

2. Abstract. Results. It is suggested to describe the total number of samples sent for pathogen detection. Among these samples, 97 was positive for HAdV and proceeded for further sequencing.

3. Methods. Diagnostic criteria of pneumonia are quite loose. 

4. Methods. Statistical analysis. (Page 9, Line 162) The Kruskal-Wallis test is designed to compare the medians of three or more independent groups. In this study, the characteristics were compared between two groups (e.g. with and without pneumonia). It is more appropriate to use the Mann-Whitney U test rather than the Kruskal-Wallis test. In addition, Kruskal-Wallis test was misspelled. 

5. Methods/Results. The major drawback of this study is the detection method. Detection of respiratory virus by PCR assay could only indicate the presence of DNA of virus and could not provide sufficient information about infectivity or acute infection. Such carryover detection of nucleic acid of inactivated viruses in certain species could result in an over-weighted effect on epidemiological analysis. 

6. Results. Another major flaw is the definition of co-infection. The results of the microbial agents identified in this study should be co-detection instead of co-infection. Particularly the diagnosis of Mycoplasma pneumoniae infection was based on the antibody test. 

7. Results. The description of the Table 1 is hard to understand. Several characteristics had mysterious denominators. It is suggested to check it carefully. 

8. Results. Table 1. It is suggested to show the neutrophil and lymphocyte counts instead of %. In addition, the unit of platelet count and CRP should be provided. 

9. Results. Figure 1 and 2 are unnecessary. All the information were shown in the Table 1. 

10. Results. Table 2. The number of patients with influenza shown here is inconsistent with the data described in the text.

11. Results. Figure 4. The numbers of tested adenovirus 3 and 7 were inconsistent with the data described in the text. 

12. There are several typos in the manuscript. The authors should check it carefully. English editing is also highly recommended.

Reviewer #2: The study stratified and grouped individuals based on the severity of infection and adenovirus genotypes during a local adenovirus outbreak. It combined clinical major symptoms and auxiliary examination results, particularly analyzing co-infection with other pathogens. The research topic and innovation are acceptable, but there are still many shortcomings.

First, it was not stated whether patients with a single adenovirus positive infection had the same severity of illness as those with multiple pathogens, since only patients with more severe disease were more likely to be tested for multiple pathogens. Second, the chart is too rough, some of the conventional notes are not in place, making it difficult to understand. In addition, the discussion section retells a lot of results, which should further deepen the significance of the research.

Reviewer #3: In this paper, Nguyen et al report about molecular typing of human adenovirus associated with an outbreak of respiratory infection among children in Northern Vietnam and risk factors for more severe cases. 

Comments for minor revision

1. In the introduction, first paragraph line 48 -50, the authors should mention more than 100 serotypes and refer to the homepage of the adenovirus working group (HAdV Working Group (gmu.edu) ). They should also mention that HAdV can cause severe or fatal cases even death in both immunocompetent and immunocompromised individuals.

2. As the study involves identifying HAdVs types, please describe how typing of HAdVs is performed in the Introduction to give the reader a clear understanding.

3. The references for HAdV outbreaks with respiratory illnesses, conjunctivitis, and gastrointestinal disorders should be include in line 51 -52 instead of bunching all the references in line 54

4. Please use the same abbreviation for the same types of HAdV e.g either use HAdV-2 or HAdV-C2 throughout the manuscript not both. See line 54 -58

5. Line 72 – there should not be a full stop between “ acute respiratory distress . which required intensive treatment and prolonged hospitalisation) replace with a comma

6. Since HAdV is a DNA virus, the viral extract would be DNA not RNA. Would recommend replacing “ partly used to extract RNA by QIAamp Viral RNA Kit “ with subjected to nucleic acid extraction using the QIAamp Viral RNA Kits within 24 hours after collection (line 96-97). Also “Extracted RNA samples “ should be replaced with Extracted nucleic acids were tested for HAdV and 6 other respiratory pathogens. 

7. In the method section line 103, can you please briefly state how the HAdV molecular typing was done i.e was it based on the hexon or fibre or penton base gene or both hexon and fibre genes. If hexon gene what region i.e 1-6 or 1-7 hypervariable region

8. Lines 124 -128 – HadV should be HAdV. Please correct throughout the manuscript

9. Lines 140 - replace “only sample with a detectable appropriate PCR product from nested-PCR “ with only samples showing amplicons with the expected band size from nested-PCR ……

10. What was the classification for the history of severe respiratory disease? This is not mentioned in the methods.

11. Line 201 – Bacterial_coinfection not “Vacterial_coinfection”. Also either use coinfection or co-infection throughout the manuscript not both

12. Although typing of HAdV based on hexon 1-6 may be sufficient for rapid identification of HAdV types during an outbreak or for routine large-scale molecular epidemiological monitoring, it is not enough to identify HAdV types according to the sequence of hypervariable regions 1-6 of the hexon gene as HAdVs are prone to recombination. please discuss the limitation of using only the hexon for HAdV typing. 

13. There is inconsistency in the number of samples that were identified as HAdV-3 and 7. The number in Table 1 for HAdVs identified as type 3 and 7 (81 and 15 samples respectively) does not match what is stated in the results section line 310 (82 samples for type 3 and 14 samples for type 7) and also in the discussion line 343. Line 310 “ Ninety-seven (98.98%) HAdV samples were genotyped successfully”, shouldn’t this be 100% of the samples instead of 98.98% as stated?

Reviewer #4: The manuscript would benefit significantly from a revision of the use of language by a native speaker of English

The INTRODUCTION section needs a brief description of the current classification of adenoviruses and a consistent nomenclature to describe HADV species and types.

The REFERENCES provided by the authors starting on line 400 are incomplete.

PLOS authors have the option to publish the peer review history of their article (what does this mean?). If published, this will include your full peer review and any attached files.

Reviewer #1: No

Reviewer #2: No

Reviewer #3: No

Reviewer #4: No
---

## [Decision Letter · Decision Letter 1]

5 Oct 2023

Dear Dr. Thi Ho,

Thank you very much for submitting your manuscript "Molecular subtypes of Adenovirus-associated acute respiratory infection outbreak in children in Northern Vietnam and risk factors of more severe cases" for consideration at PLOS Neglected Tropical Diseases. As with all papers reviewed by the journal, your manuscript was reviewed by members of the editorial board and by several independent reviewers. The reviewers appreciated the attention to an important topic. Based on the reviews, we are likely to accept this manuscript for publication, providing that you modify the manuscript according to the review recommendations. 

Sincerely,

Chukwunonso Nzelu, Ph.D.

Guest Editor

Elvina Viennet

Section Editor

Reviewer's Responses to Questions

**Key Review Criteria Required for Acceptance?**

**Methods**

-Are the objectives of the study clearly articulated with a clear testable hypothesis stated?

-Is the study design appropriate to address the stated objectives?

-Is the population clearly described and appropriate for the hypothesis being tested?

-Is the sample size sufficient to ensure adequate power to address the hypothesis being tested?

-Were correct statistical analysis used to support conclusions?

-Are there concerns about ethical or regulatory requirements being met?

Reviewer #1: (No Response)

**Results**

-Does the analysis presented match the analysis plan?

-Are the results clearly and completely presented?

-Are the figures (Tables, Images) of sufficient quality for clarity?

Reviewer #1: (No Response)

**Conclusions**

-Are the conclusions supported by the data presented?

-Are the limitations of analysis clearly described?

-Do the authors discuss how these data can be helpful to advance our understanding of the topic under study?

-Is public health relevance addressed?

Reviewer #1: Yes

**Editorial and Data Presentation Modifications?**

Reviewer #1: Minor Revision

**Summary and General Comments**

Reviewer #1: Several bacterial species names are misspelled in the text and footnote of tables. For example, Staphylococcus aureus, Legionella pneumophila, etc. The authors should check it carefully throughout the manuscript.

PLOS authors have the option to publish the peer review history of their article (what does this mean?). If published, this will include your full peer review and any attached files.

Reviewer #1: No

Figure Files:

Data Requirements:

Reproducibility:

References

---

## [Editor Report · Decision Letter 2]

19 Oct 2023

Dear Dr. Ho,

We are pleased to inform you that your manuscript 'Molecular subtypes of Adenovirus-associated acute respiratory infection outbreak in children in Northern Vietnam and risk factors of more severe cases' has been provisionally accepted for publication in PLOS Neglected Tropical Diseases.

Best regards,

Chukwunonso Nzelu, Ph.D.

Guest Editor

Elvina Viennet

Section Editor

---

## [Editor Report · Acceptance letter]

31 Oct 2023

Dear Dr. Ho,

We are delighted to inform you that your manuscript, "Molecular subtypes of Adenovirus-associated acute respiratory infection outbreak in children in Northern Vietnam and risk factors of more severe cases," has been formally accepted for publication in PLOS Neglected Tropical Diseases.

Best regards,

Shaden Kamhawi

co-Editor-in-Chief

Paul Brindley

co-Editor-in-Chief
